# Hydrogen Attenuates Inflammation by Inducing Early M2 Macrophage Polarization in Skin Wound Healing

**DOI:** 10.3390/ph16060885

**Published:** 2023-06-15

**Authors:** Pengxiang Zhao, Zisong Cai, Xujuan Zhang, Mengyu Liu, Fei Xie, Ziyi Liu, Shidong Lu, Xuemei Ma

**Affiliations:** 1Faculty of Environment and Life, Beijing University of Technology, Beijing 100124, China; zpx@bjut.edu.cn (P.Z.);; 2Beijing Molecular Hydrogen Research Center, Beijing 100124, China; 3Beijing International Science and Technology Cooperation Base of Antivirus Drug, Beijing 100124, China

**Keywords:** molecular hydrogen, wound healing, inflammation stage, M2 macrophage polarization, in vivo time series study, anti-ROS independent

## Abstract

The heterogeneous and highly plastic cell populations of macrophages are important mediators of cellular responses during all stages of wound healing, especially in the inflammatory stage. Molecular hydrogen (H_2_), which has potent antioxidant and anti-inflammatory effects, has been shown to promote M2 polarization in injury and disease. However, more in vivo time series studies of the role of M1-to-M2 polarization in wound healing are needed. In the current study, we performed time series experiments on a dorsal full-thickness skin defect mouse model in the inflammatory stage to examine the effects of H_2_ inhalation. Our results revealed that H_2_ could promote very early M1-to-M2 polarization (on days 2–3 post wounding, 2–3 days earlier than in conventional wound healing), without disturbing the functions of the M1 phenotype. Time series analysis of the transcriptome, blood cell counts, and multiple cytokines further indicated that peripheral blood monocytes were a source of H_2_-induced M2 macrophages and that the functions of H_2_ in macrophage polarization were not only dependent on its antioxidant effects. Therefore, we believe that H_2_ could reduce inflammation in wound care by shifting early macrophage polarization in clinical settings.

## 1. Introduction

Wound healing is normally comprises four overlapping but distinct stages: hemostasis, inflammation, proliferation, and remodeling [1]. Failure at any stage may lead to the development of chronic non-healing wounds or excessive scar formation. Skin wound healing is a highly coordinated process involving multiple tissue-resident and recruited cell types, as well as cytokines and growth factors [2]. Among all these participants, macrophages play a critical role, first in killing potential pathogens (innate immune defense) and then suppressing excessive inflammation, which is essential for tissue remodeling and regeneration [3,4]. In all healing stages, macrophages can either originate from skin tissue-resident cells or develop from monocytes [5]. According to the skin architecture, two types of macrophages reside in the skin tissue: epidermal Langerhans cells and dermal macrophages [5]. Both populations can be replenished by blood monocytes in the inflammatory stage [6]. Bone marrow-derived monocytes are very important for the replenishment of tissue-resident macrophages [7]. Although the mechanisms underlying the activation and proliferation of skin-resident macrophages remain unclear [8], the plasticity of macrophages, which has been proven to influence outcomes in wound healing, has drawn much interest [9,10].

The heterogeneous and highly plastic cell populations of macrophages are considered important mediators for cellular responses during all stages of wound healing [11]. Generally, macrophages are divided into two subtypes: the pro-inflammatory (classically activated) M1 phenotype and the anti-inflammatory (alternatively activated) M2 phenotype [12]. An even broader spectrum of macrophage phenotypes has been found in physiologic and pathologic conditions in vivo [11,13,14]. Conventionally, M1 macrophages predominate from day 1 to day 3 post wounding, cooperating with other wound-healing cells [15,16]. The transition from M1 to M2 usually occurs after the early inflammatory stage, peaking around day 7 [17]. The proportion of macrophage subtypes shifts from about 85% M1 on days 1–3 to 80–85% M2 on days 5–7 [14]. However, the dynamic proportions of macrophage phenotypes also depend on the wound microenvironment [7]. M1 polarization is evoked by TLR ligands such as LPS, IFN-γ, and TNF [18]. On the contrary, M2 polarization is driven by cytokines such as IL-4, IL-10, and IL-13 [18]. The signaling cascades involved include the interferon-responsive factor (IRF)/STAT pathway [19,20,21]. The M1/M2 balance strongly influences tissue inflammation and injury repair [22]. Cytokine production by the two macrophage subtypes differs between the early and late inflammatory stages [23]: M1 macrophages mainly produce pro-inflammatory mediators including TNF-α, IL-1β, IFN-γ, IL-6, IL-12, IL-23, iNOS, and reactive oxygen species (ROS), while M2 macrophages tend to produce more pro-angiogenic and pro-fibrotic factors such as IL-10, TGF-β, insulin-like growth factor (IGF), epidermal growth factor (EGF), and vascular endothelial growth factor (VEGF), which further contribute to wound closure and tissue repair [24]. Though M2 macrophages exert pro-healing effects, the effects of M1 macrophages should also not be overlooked. Therefore, on the premise of not affecting the pro-inflammatory function of M1-macrophages, shortening the inflammatory period by advancing M1-to-M2 polarization may be a key strategy by which to promote wound healing.

Molecular hydrogen (H_2_) is known to exert potent antioxidant and anti-inflammatory effects in diverse diseases [25,26,27,28] by modulating the activities and expression of molecules such as Lyn, ERK, p38, iNOS, NF-κB, and STAT3. Studies have also revealed that H_2_ therapy promotes wound healing in different skin injury disease models, including burn wounds [29], pressure ulcers [30,31], diabetic wounds [32,33], and radiation-induced dermatitis [34], indicating that H_2_ treatment is a promising strategy in clinical wound care. Moreover, H_2_ has been shown to shift macrophage polarization in tissue injury repair. In studies of acute lung injury, hydrogen-rich saline exerted a protective effect by decreasing autophagy and apoptosis, and inducing M1-to-M2 polarization in vivo and in vitro after LPS-induced lung injury [35]. In a study of bleomycin-induced lung injury, H_2_ inhalation (3.2% in air) was used as daily treatment, and a significantly higher proportion of M2 macrophages was observed in the alveolar interstitium in the H_2_ group 7 days after injury [36]. Ischemic stroke is also a kind of inflammatory disorder in which microglia and macrophages in the central nervous system play important roles in regulating the inflammation responses [37]. Ning et al. [38] conducted an in vitro time-course analysis using RAW264.7 macrophages and reported that H_2_ significantly decreased the proportion of M1 macrophages on day 3 after LPS treatment, which lasted until day 5. No influence on M2 polarization was observed [38]. Their results were not entirely consistent with another study of acute kidney injury (also in RAW264.7 cells) [39], which showed that the inhalation of hydrogen-rich solution attenuated M1 and increased M2 polarization both in vivo and in vitro. Some researchers also pointed out that PMA-differentiated THP-1 cells were morphologically and functionally more similar to human peripheral blood monocytes, suggesting that they are more suitable in the study of relatively straightforward biological processes such as in vitro polarization [40]. In a recent study of H_2_ storage materials, a spontaneous H_2_-releasing nanosheet was shown to have antioxidant effects, downregulating the RNA levels of inflammatory cytokines, reducing the proportion of M1 macrophages, and increasing the proportion of M2 macrophages in vitro [41], indicating that it could be used in osteoarthritis treatment.

When conducting in vitro experiments of macrophage polarization, it is very difficult to mimic the complex in vivo microenvironment. Therefore, more information on H_2_-induced macrophage polarization in cutaneous wound healing is needed, especially during the early inflammatory stage. In addition, the mechanisms underlying macrophage polarization, including the key genes involved, need to be investigated. In a recent study, we showed that H_2_ significantly accelerated wound healing by inducing the early deposition of the extracellular matrix and self-epidermal stem cell proliferation [42]. Hydrogen induces the M1-to-M2 polarization of macrophages, but recent research has found it in the late healing period. In our study, the earliest onset of polarization was detected at the beginning of inflammation. Early temporal studies have been conducted in vitro. M1-to-M2 polarization events in early inflammation in vivo have not yet been revealed. Our study fills this gap at the transcriptomic, cytokine, and histological levels.

In this study, a time series experiment was performed on a dorsal full-thickness skin defect model during the inflammatory stage. We revealed that H_2_ could promote early M1-to-M2 macrophage polarization (day 3 post wounding), without disturbing the functions of M1 macrophages, which was anti-oxidant independent. Time series analyses of the transcriptome and multiple cytokines further indicated that H_2_ advanced the M1-to-M2 transition, which started 2–3 days earlier than in ordinary wound healing processes. Therefore, H_2_ could reduce inflammation in wound healing by regulating the very early macrophage polarization, which provided experimental evidence for the clinical treatment of hydrogen intervention in early wound healing.

## 2. Results

### 2.1. H_2_ Reduced the Inflammatory Response of Skin Wounds

#### 2.1.1. H_2_ Promoted Early Wound Healing and Reduced Inflammatory Cell Infiltration with Better Effects Than NAC

To investigate the active role played by H_2_ in the inflammatory phase of skin wound healing, we established a mouse model of a total dorsal skin defect [43]. Most studies have suggested that H_2_ exerts its anti-inflammatory effects by targeting ROS; therefore, in the present study, we used the classical antioxidant N-acetyl-L-cysteine (NAC) as a control, which reduces ROS levels in cells and attenuates the abnormal peroxidative response of the organism. The H_2_ inhalation device, experimental period, sampling region, and scheme of the wound are shown in Figure 1A–C. H_2_ has been reported to promote wound healing, with healing effects observed as early as day 3 [44]. Our experimental results were consistent with this finding above, with a higher degree of wound healing, a pinker wound bed, and the formation of fewer blood clots within 3 days in the H_2_ treatment group compared to the NAC and control groups, suggesting a lower degree of inflammation (Figure 1D).

The occurrence of H_2_-promoted wound healing signifies diminished inflammation, and may be accompanied by a decrease in inflammatory cell infiltration at the wound site and a decrease in inflammatory cells in the blood. We observed the degree of inflammatory cell infiltration via the H&E staining of proximal skin wound tissue on days 1–3 (Figure 1E), compared with non-damaged skin (Figure 1F) in the proximal wound. The H_2_ treatment group showed less infiltration of inflammatory cells in the skin dermis than the other two groups from 24 to 72 h.

#### 2.1.2. H_2_ Reduced Inflammatory Cell Infiltration at the Wound Site at an Early Stage of Wound Healing

The H&E staining results showed that the hydrogen treatment group saw a reduction in the infiltration of inflammatory cells in the proximal wound tissue on days 1–3. Therefore, we confirmed this conclusion via IHC staining of the proximal skin wound tissue on days 1–3 (Figure 2), counting the number of T-lymphocytes (CD3) and B-lymphocytes (CD19) in the proximal wound, (*p* < 0.05 between the H_2_ group and the control group at 24 h). The results showed that the number of T lymphocytes (CD3) in the hydrogen treatment group significantly decreased compared to the control group at 48 and 72 h (*p* < 0.01), while the NAC treatment group significantly decreased compared to the control group at 48 h (*p* < 0.01). B-lymphocytes (CD19) in the hydrogen treatment group significantly decreased compared to the control group at 24 and 72 h (*p* < 0.01).

### 2.2. H_2_ Promoted the Polarization of M2-Type Macrophages during the Inflammatory Phase

#### H_2_-Induced In Vivo Time Series M2-Type Macrophage Polarization at the Early Inflammatory Stage of Wound Healing

Macrophages can colonize from the peripheral monocytes in the wound sites and polarize into pro-inflammatory M1-type macrophages and anti-inflammatory M2-type macrophages. It is well known that the M1/M2 balance affects the fate of an organ or tissue in inflammation or injury [23]. Considering the slightly decreased monocyte number in the H_2_ group at 72 h, we manually counted and evaluated the polarization of M1 and M2-type macrophages in proximal skin wound tissue at 24, 48, and 72 h (Figure 3A). The H_2_ group had a progressively lower proportion of M1-type macrophages in the wound from 24 h to 72 h (Figure 3B); the proportion of M1-type macrophages was lower in the H_2_ group than in the control group at 48 h (*p* = 0.058 between the H_2_ group and the control group, i.e., not statistically significant) and at 72 h (*p* < 0.05); and the M1-type macrophage proportion was even lower in the H_2_ group than in the NAC group. The proportion of M1-type macrophages was significantly lower in the NAC group than in the control group at 48 h (*p* < 0.05) (Figure 3B), but reached a similar level at 72 h between the two groups (Figure 3B). The proportion of M2 macrophages gradually increased in the H_2_ and NAC groups from 24 to 48 h compared with the control group. At 72 h, the proportion of M2 macrophages was significantly higher in the H_2_ group than in the control group (*p* < 0.05) and higher than in the NAC group (Figure 3B). The M2/M1 ratio was higher in the H_2_ and NAC groups than in the control group at 48 h, and at 72 h, this ratio was significantly higher in the H_2_ group than in the control group (*p* < 0.05), and almost twice that in the NAC group (Figure 3C). Thus, H_2_ treatment induced early M2 polarization in the wound tissue, which was more efficient than NAC.

In order to quantitatively and objectively analyze the ratio of M1/M2 macrophages in tissues, quantitative real-time PCR (qRT-PCR) was used to determine the expression levels of the M1 macrophage marker CD86 and the M2 macrophage marker CD163 in H_2_, NAC, and control groups at 1 to 3 days (Figure 3D). The results showed that the expression of the CD86 gene in the H_2_ and NAC groups was lower than that in the control group at 48 h, but the difference was not significant, and that its expression in the H_2_ group was significantly lower than that in the control group at 72 h (*p* < 0.05). The expression of the CD163 gene in the H_2_ group was significantly higher than that in the control group at 72 h (*p* < 0.05).

An in vitro cell experiment was performed to confirm the effect and direction of H_2_ on the polarization of M2-type macrophages. The differentiation of THP-1 cells into M0-type macrophages was induced with PMA, followed by the induction of the polarization of M0-type macrophages with H_2_-containing medium. LPS+IFN-γ-induced M1-type macrophage polarization and IL-4+IL-13-induced M2-type macrophage polarization were used as controls (Figure 3E). The results revealed that H_2_ could induce M2 polarization (Figure 3F). In addition, we confirmed, by using oxygen electrode, that the oxygen content in the hydrogen-saturated medium was basically the same as that in the control medium. The relevant results are attached.

### 2.3. Transcriptome Time Series Cibersort Immune Infiltration Analysis Confirmed H_2_ Functions on M2 Macrophage Polarization

Subsequently, Cibersort immune infiltration analysis [45] was performed on transcriptomic data from the first three days of skin wound samples (Figure 4). The numbers of T cells, B cells, and other relevant immune cells in each group were similar across groups, indicating that the wound was in a sterile microenvironment (see raw data in Appendix A). To investigate the effects of H_2_ on macrophages, monocytes (differentiated into macrophages), M1 macrophages, and M2 macrophages were counted (Figure 4, right). The results were generally in agreement with the results of our tissue immunofluorescence experiments and in vitro assays, with monocytes being more numerous in the H_2_ group than in the control group at 48–72 h, possibly promoting their colonization of the wound bed. Meanwhile, at 72 h, H_2_ was confirmed to increase the polarization of M2 macrophages (Figure 4C).

### 2.4. H_2_ Promotes the Expression of M1/M2-Type Macrophage-Related Factors and Time Series of Heatmap Gene Clustering Showing H_2_-Induced Early M2-Type Macrophage Polarization

The expressions of macrophage-related cytokines, cell surface markers, and transcription factors usually reflect the degree of M1 and M2 polarization. The time series of the detection of macrophage-related factors were processed. Firstly, we examined the serological levels of the pro-inflammatory factors IL-1β, TNF-α, IL-6, IL-17A, and IFN-γ, as well as the anti-inflammatory factor IL-10 (Figure 5). IL-1β, TNF-α, IL-6, and IFN-γ are the main cytokines secreted by M1-type macrophages. Overall, the expression of these three cytokines in the H_2_ treatment group gradually decreased from 24 to 72 h, and their expression levels were all lower than those in the control and NAC groups at 72 h (Figure 5B,C,E), with IFN-γ levels already lower than those in the NAC and control groups at 48 h and significantly different compared to the NAC group at 72 h (*p* < 0.05) (Figure 5B). The levels of IL-1β and IL-17A (Figure 5), which are presumably involved in normal inflammatory regulation, were not significantly different between the groups over three days. All the results above indicate that H_2_ did not disturb the functions of M1-type macrophages during the very early stages of inflammation. M2-type macrophages mainly secrete anti-inflammatory cytokines, such as IL-4, IL-10, and IL-13, to alleviate the inflammatory response at the wound site. According to our data, the expression of IL-10 in the H_2_ and NAC groups was higher than that in the control group at 48 and 72 h, and IL-10 expression in the H_2_ group was the highest among the three groups at 72 h. The removal of ROS (NAC group) may also contribute to the polarization of M2-type macrophages, but to a lesser extent than the induction of H_2_.

Further, the expression levels of M1/M2 polarization-related genes were selected and checked from the RNAseq data for proximal skin wound tissue in the H_2_ and control groups on days 1–3 post wounding (Appendix A). Heatmap gene clustering revealed M1-type macrophage-related genes, including genes coding secretory proteins (IFN-γ, IL-1β, IL-1α, IL-12α), surface-expressed proteins (CD14, CD32, CD68, CD80, CD86, CD204, CD369), and transcription factors (Mer, MCHII, IRF5) (Figure 5G). The results showed that the expression of M1-type-related genes in the H_2_ group was higher than that in the control group at 48 h and lower than that in the control group at 72 h (Figure 5G). The expression of most M2 macrophage-related genes was higher in the H_2_ treatment group than in the control group at 24 and 48 h, including genes encoding secreted proteins (Arg1, IDO, IL-4, IL-10, TGF-β, YM1), surface-expressed proteins (CD14, CD115, CD163, CD206), transcription factors (CDF1R, LY-6C, IRF4, STAT6), and M2-type-related genes (Figure 5H). The results showed that H_2_ maintained the M1-type macrophage polarization process on the first two days, and accelerated the early M1-to-M2 polarization.

## 3. Discussion

Considering the anti-inflammatory and antioxidant functions of H_2_, it is very important to reveal the early events of the inflammatory stage during the wound healing process. In vivo time series observations of H_2_-induced macrophage polarization during wound healing are often lacking, particularly in the early phase of healing processes. In our study, in contrast to the typical macrophage phenotypic transition in wound healing, H_2_ treatment significantly accelerated the M1-to-M2 transformation (at least 2–3 days earlier). Immunofluorescent staining results showed that in vivo M1-to-M2 macrophage polarization started on day 2 post wounding at the proximal wound (Figure 3A–C). The M2 proportion reached around 70% on day 3 in the H_2_ group, while it was less than 40% in the control group (Figure 3B). On the contrary, the M1 proportion in the H_2_ group on day 3 declined to half of the control group (Figure 3B,C). This result was confirmed by qRT-PCR (Figure 3D). Time series analysis of the immune cell phenotypes (24 h, 48 h, and 72 h) corroborated these results, indicating that M2 polarization in the H_2_ group exceeded that in the control group on day 3 post wounding (Figure 4).

Skin wound macrophages comprise tissue-resident macrophages and infiltrating monocytes recruited from peripheral blood. Both are crucial to the inflammatory stage of wound healing [46]. In the present study, infiltrating monocytes could be one of the sources of M1 and M2 macrophages in the H_2_ group. In the H_2_ group, as opposed to the control group, the monocyte numbers in the whole blood on days 2–3 post wounding decreased (Appendix A); At 72 h, the number of M2 macrophages in the proximal wound tissue was higher than that in the control group (Figure 4C). In the NAC group, the number of peripheral monocytes on days 2–3 post wounding decreased in the same way as in the H_2_ group (Appendix A); however, the M2 macrophage population in the NAC group on day 3 lagged behind that in the H_2_ group, indicating a weaker tendency to polarize toward the M2 phenotype under NAC treatment. Nevertheless, more studies are required to determine whether H_2_ increases M2 polarization in tissue-resident macrophages.

Although the two dominant macrophage subtypes are simply characterized as pro-inflammatory (M1) and anti-inflammatory (M2), we should keep in mind that both M1 and M2 macrophages are essential for successful wound healing. In particular, M1 macrophages, which are responsible for driving the inflammatory response, become more numerous during the first phases of wound healing. M1 macrophages produce ROS and other inflammatory cytokines, which are part of the normal inflammatory process and trigger the following stages of wound healing [47]. In addition, ROS are critical for M2 macrophage transformation; for example, in cancer mouse models, antioxidant treatment prevents the differentiation of tumor-associated macrophages (M2-like macrophage) and carcinogenesis [48]. Therefore, the inappropriate removal of ROS and the inhibition of inflammatory responses may hinder the wound healing process. In the present study, H_2_ inhalation showed moderate anti-inflammatory effects.

Depending on the distinct directions of macrophage polarization, the cytokine profile [49,50] and markers change accordingly [51,52]. Serological cytokine profiling in the H_2_ group revealed minor variations over the first two days after injury, as well as a decline in IFN-γ and other Th1- and Th17-related cytokines on day 3. The anti-inflammatory Treg-related cytokine IL-10 was elevated on day 3 in the H_2_ group. Gene expression profiling on the first three days suggested that both M1 and M2 cell surface markers were upregulated in the H_2_ group during the first two days, which may indicate an early transition phase between M1 and M2 under H_2_ treatment. On the third day, M1 marker genes predominated in the control group, while the M2 marker genes showed similar expression levels between the two groups. Among these M1- and M2-related genes, genes encoding TGF-β and the transcription factor STAT6 (Figure 5G,H) could be two of the key genes, providing insight into the signaling pathways involved in H_2_-induced M2 polarization.

For a very long time, it was widely accepted that H_2_ has cytoprotective effects against oxidative stress and functions as a therapeutic and preventative antioxidant by selectively eliminating highly active oxidants such as hydroxyl radicals and peroxynitrite [25]. Tissues are damaged when reperfusion begins due to excessive oxidative stress, and early phases of the healing process for cutaneous wounds cause oxidative stress. According to our study, H_2_ does not primarily promote wound healing via these antioxidant mechanisms. To prove this, the ROS scavenger NAC was used as a control [53]. NAC has been reported to promote prostate wound healing by removing ROS and ROS-producing M1 macrophages [54]. Although we observed the beneficial effects of NAC treatment, for example, on wound closure (Figure 1D), the IHC results show early infiltration numbers of lymphocytes (Figure 2), and the early tendency of M1/M2 to transform (Figure 4); these effects were not as good as in the H_2_ group. Specifically, the M1 proportion in the NAC group on day 1 (Figure 3B) was higher and the M2 proportion was lower than in the H_2_ group (Figure 3B), resulting in a greater M2/M1 ratio in the NAC group (Figure 3C). On days 2 and 3, the M2 proportion was higher in the H_2_ group, causing the first three days’ healing rates in the H_2_ group to be faster than in the NAC group and the control group.

One of the limitations of our study is related to the cause of different macrophage polarization outcomes between the H_2_ and NAC groups. A possible explanation could be the characteristics of the selective oxidation resistance of H_2_, which, according to the research from Ohta [25], decreases •OH levels but has no effect on •O_2_, H_2_O_2_, or •NO levels, which have multiple physiological functions [42]. NAC acts both as a direct antioxidant and as a glutathione (GSH) precursor. GSH is a sacrificial scavenger against harmful ROS, including R-OOH and •O_2_^−^. NAC is able to react with •OH, •CO_3_^−^, •NO_2_, and HOCl. As a result, NAC treatment may not be as successful as treatment with H_2_. Another hypothesis is that H_2_ has the smallest molecular weight among all molecules, facilitating the perfusion of any tissue under any physiological and pathological condition. This diffusion is not dependent on blood flow, enabling it to reach the wound tissue more easily and exert beneficial functions than other antioxidants. In addition, there is still a possibility that H_2_ affects the activity of different cells. According to our latest research, in the early stages of wound healing, H_2_ can affect the activity of multiple cells, the accumulation of extracellular matrix, and even the proliferation and differentiation of stem cells, thus promoting rapid wound healing. More macrophage lineage tracing studies under different conditions are required to determine how H_2_ impacts the cell fate of macrophages. 

## 4. Materials and Methods

### 4.1. Ethics Statement

All animal studies were conducted in accordance with the protocols approved by the Biomedical Research Ethics Committee of the Sixth Medical Center of PLAGH, China, and all procedures were conducted in accordance with the Regulations on the Administration of Experimental Animal Affairs (China). The ethical approval number: HZKY-PJ-2021-38. 

### 4.2. Animal Model and Grouping

Forty-five male C57BL/6J mice (Viton Lever Laboratory Animal Technology Co., Ltd., Beijing, China), 7 weeks old and weighing 20–24 g were used in this study. The animals were maintained in standard conditions at 22–25 °C and were fed and watered ad libitum under a 12 h/12 h light/dark cycle. Enrofloxacin (0.17 mg/mL) was added to the drinking water daily. Mice were allowed to acclimatize to their environment for one week prior to the experiment.

A full-thickness cutaneous wound healing model in mice was established (Figure 1C). The animals were anesthetized with (20 mg/mL, 10 μL/g) tribromoethanol, and their backs were shaved and cleaned with 75% ethanol. We sutured the shaved dorsum with a concentric circular silicone membrane (to prevent wound closure due to the contraction of the mouse dorsal skin), and used sterile scissors to cut the entire skin layer 10 mm in diameter in the center of the gasket (Figure 1C,D).

The animals were randomly divided into three groups (each group contained 15 mice, and 5 mice from each group were sacrificed every 24 h at three time points). Group H_2_ was placed in the chamber containing the air mixture for 2 h per day for three days, For the NAC group, mice were injected with 150 mg/kg of NAC (the ROS scavenger acetylcysteine; #S1623, Selleckchem, Shanghai, China) in the 22 (±2) g body weight range.

### 4.3. H_2_ Inhalation Chamber Preparation

A mixed H_2_ generator (KLE H7, Shenzhen, China) generating 66% H_2_ and 33% O_2_ was connected to a transparent airtight box attached to the top (Figure 1A). The animals were placed in the box containing the air mixture for 2 h each day. Each group was administered the treatment once daily until the end of the experiment (Figure 1B). During this time, the mice were allowed to move freely.

### 4.4. Cell Culture

Human mononuclear leukemia cells (THP-1) (MBCR,10th generation, G10) were routinely cultured in PRMI1640 medium (Gibco, New York, NY, USA) supplemented with 10% fetal bovine serum (Gibco, New York, NY, USA) and 1% penicillin and streptomycin (Gibco, New York, NY, USA) at 5% CO_2_ and 37 °C.

### 4.5. H_2_ Rich Medium Preparation

The H_2_-enriched medium was prepared by passing the gas generated by the H_2_ generator into the PRMI1640 medium for 20 min. The H_2_ concentration dissolved in the medium was detected using a H_2_ electrode (Unisense, Aarhusn N, Denmark) following the method of a previous article.

### 4.6. Hematoxylin and Eosin Staining (H&E) Immunohistochemistry (IHC) Immunofluorescence (IF) Staining for Paraffin Slides

Skin tissue from the wound site was embedded in paraffin, made into 5 μm thick sections, and fixed on glass slides for H&E staining. For immunohistochemistry, tissue sections were defatted and hydrated in xylene, anhydrous ethanol (100%, 95%, 70%, 50%), subsequent antigen repair, goat serum for blocking, and incubated overnight at 4 °C with the specific primary antibodies CD3 (Proteintech,60181-1-ig) and CD19 (Proteintech, 27949-1-AP). Immunofluorescence staining, the Pathological sections were incubated overnight at 4 °C with specific primary antibodies CD86 (Invitrogen, 14086282) and CD163 (Invitrogen, PA578961), respectively. Further labeling was performed with specific secondary antibodies. The fixed cells were stained with NucBlue Fixed Cell ReadyProbes (Invitrogen, R37606) dye for nuclei. 

Images of the samples were obtained under a microscope, with each section taken at 10× and 40× magnification, respectively. The specifically labeled M1 macrophages (CD86, red) and M2 macrophages (CD163, green) were counted, and the percentages of M1 and M2 in the total number were calculated, respectively, as well as the relative ratio of M2/M1.

### 4.7. Quantitative Real-Time PCR

RNA was extracted from skin wound tissue. The cDNA was synthesized using a ReverTra Ace qPCR-RT Master Mix cDNA Synthesis Kit (TOYOBO, FSQ-201). The integrity and concentration of the cDNA were measured using a NanoDrop 2000 machine (Thermo Scientific, Waltham, MA, USA). The expressions of the target genes were evaluated via qRT-PCT on 700 Fast Real-Time PCR Systems (ViiA7 Real-time PCR, ABI), using a AceQ qPCR SYBR Green Master Mix Kit (Vazyme Biotech, Nanjing, China). The DNA primers used were mouse CD86 Forward: 5′ ATATGACCGTTGTGTGTGTTCTGGA 3′ and reverse 5′ AGGGCCACAGTAACTGAAGCTGTAA 3′: CD163 Forward: 5′ GCCAAACCGTGGAGTCACAG 3′ and reverse 5′ GGACCAATAGAATGGCTCCACAA 3′; GAPDH Forward: 5′ AGGTCGGTGTGAACGGATTTG 3′ and reverse 5′ TGTAGACCATGTAGTTGAGGTCA 3′: The standard PCR cycle parameters were as follows: 95 °C for 300 s, followed by 40 cycles of 95 °C for 10 s, and then 60 °C for 30 s. The relative expression levels of mRNAs were determined via a comparative Ct (ΔΔ Ct) method.

### 4.8. In Vitro Macrophage Culture and Polarization

The THP-1 cell suspension was spread evenly in 48-well plates at 1 × 10^5^ cells/well. M0 macrophages were obtained by treating THP-1 cells with PMA (MedChemExpress, HY-18739) (100 ng/mL) for 48 h. Then, M0 macrophages were induced with LPS (MedChemExpress, HY-D1056) (100 ng/mL) and IFN-γ (Novoprotein, CI57) (20 ng/mL) for 48 h in the presence of PMA to acquire the M1 phenotype. Meanwhile, M0 macrophages were induced with IL-4 (Novoprotein, CX03) (20 ng/mL) and IL-13 (Novoprotein, CC89) (20 ng/mL) for 48 h to acquire the M2 phenotype. For the H_2_ group, H_2_-rich PRIM1640 medium was added into each well with M0 macrophages.

### 4.9. Immunofluorescence Staining for Cells

Immediately at the end of polarization in the last section, the wells were washed with 1 × PBS and fixed with 4% paraformaldehyde for 30 min; then, the wells were washed, the membrane was permeabilized with 0.3% Tween PBS- for 15 min, the wells were washed, and the wells were blocked with goat serum for 1 h at room temperature. Antibodies were added at the concentrations recommended by the reagent vendor and incubated at 4 °C for 24 h. The primary antibody was washed, and the appropriate secondary antibody was added for 2 h at room temperature. Finally, NucBlue Fixed Cell ReadyProbes (Invitrogen, R37606) were used to stain the nuclei of the cells.

### 4.10. Whole Blood Routine Test

Murine venous blood was collected from the eyelid. Half of the blood was diluted to a proper concentration and then the whole-blood routine test was performed by the hematology analyzer (MEK-6400, Nihon Kohden, Tokyo, Japan) at the School of Medicine, Peking University.

### 4.11. Enzyme-Linked Immunosorbent Assay (ELISA)

Murine serum was carefully collected from the other half of the whole blood after a routine test and stored at −80 °C. Serum samples underwent multiple cytokine detection by using the Bio-plex pro-TM Mouse Cytokine Th17 Panel A 6-Plex kit (#M6000007NY, Bio Rad, Hercules, CA, USA).

### 4.12. Transcriptomic Heatmaps Analysis

Transcriptomic data were obtained from the data in the previous study [42], and the M1/M2-related genes were manually selected and examined using the heatmap package in R language (v4.1.3).

### 4.13. CIBERSORT Immune Cell Infiltration Analysis

Data from the same source were analyzed and plotted using R software (version 4.1.1). CIBERSORT is currently the most cited tool for the analysis of immune cell infiltration estimates. It provides a gene expression signature set for 22 immune cell subtypes based on a known reference dataset. On this basis, the CIBERSORT deconvolution algorithm calculates the relative proportions for the immune cell infiltration analysis of the wound tissue transcriptome gene set [45]. Each sample was screened for the CIBERSORT value of *p* < 0.05 to obtain the percentage of 22 immune cells, including macrophages, monocytes, and NK cells.

### 4.14. Statistical Analysis

All statistical analyses were performed using GraphPad Prism 8 software. One or Two-way analysis of variance (ANOVA) was used for the comparison of differences between multiple groups. *p* < 0.05 was considered a statistically significant difference.

## 5. Conclusions

H_2_ has increased in significance as a potent anti-inflammation treatment in the field of wound care. Further proof that H_2_ has a role in wound healing, particularly in the early stages of the healing process, is required. Our attention has been drawn to the role of H_2_ in macrophage polarization during the inflammatory stage, which also affects the subsequent proliferative and tissue remodeling phases. According to in vivo time series analyses on the first three days after wounding, H_2_ stimulates M1-to-M2 polarization from day 1 post wounding, accelerating the process by 2–3 days without disrupting essential M1 macrophage activities (Figure 6). We also discovered that peripheral blood monocytes are one of the sources of M2 macrophages. H_2_ has anti-inflammatory and healing-promoting properties that go beyond merely ROS scavenging. Our in vivo time series analyses provide evidence that elucidates the roles of H_2_ in early macrophage polarization and that can be applied in wound healing. Further research is still needed to determine how H_2_ induces polarization and to track macrophage lineages.

## Figures and Tables

**Figure 1 pharmaceuticals-16-00885-f001:**
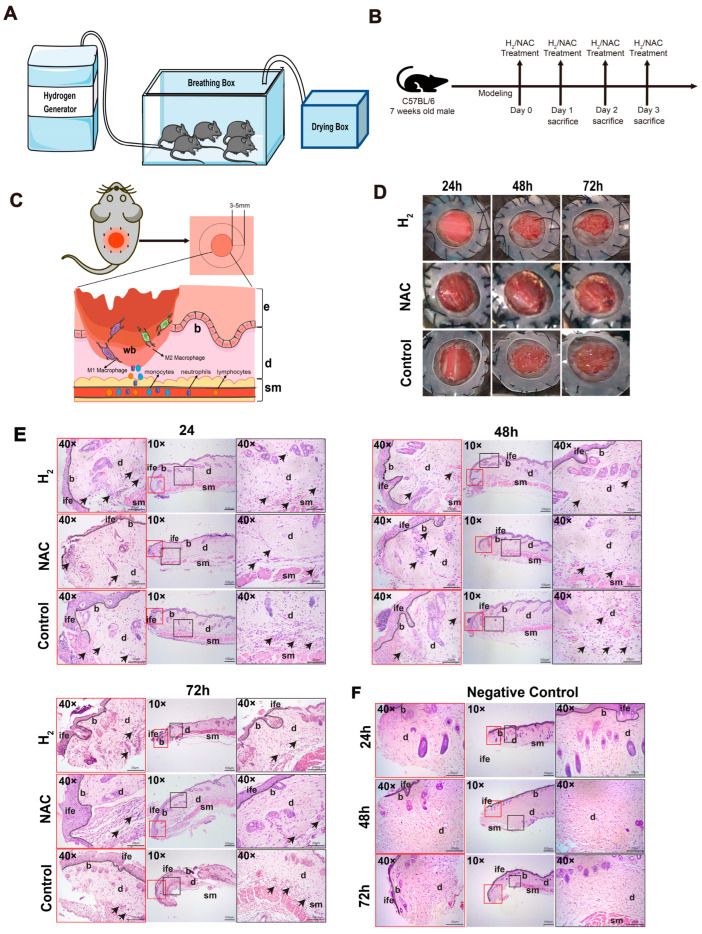
H_2_ inhalation promoted wound healing in the early stages. H&E staining revealed inflammatory cell infiltration in the proximal skin tissue during the first three days after wounding. (**A**) The components of the H_2_ inhalation chamber modules are linked to ensure that the experimental animals are in a stable H_2_ environment. (**B**) Animal experiments were performed based on a daily treatment schedule: mice immediately after modeling were treated with H_2_ and NAC once daily at 24 h intervals until sacrifice. (**C**) Up: Animals with full skin defect models were established, and skin tissue 3–5 mm from the wound was taken for subsequent experiments (the skin tissue close to the wound is labeled as the proximal end). Down: Overview of skin wounds and inflammatory cell infiltration in the presence of M1 and M2 macrophages in the wound bed is presented. (**D**) Wound closure status in the three groups during the three-day inflammatory phase. **wb**: wound bed, **b**: basal layer, **d**: dermis, **sm**: smooth muscle, **ife**: epidermis. (**E**) H&E staining results. Black arrows indicate infiltrating inflammatory cells. Magnification is indicated in the figure. (**F**) Negative control (No damage control).

**Figure 2 pharmaceuticals-16-00885-f002:**
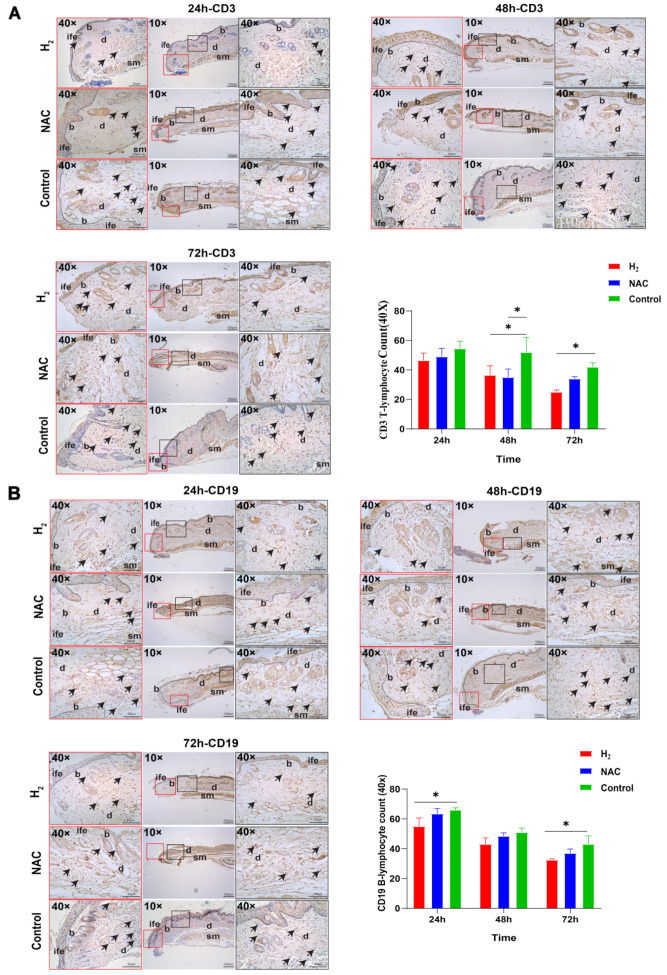
IHC staining revealed T-lymphocyte (CD3) and B-lymphocyte (CD19) infiltration in the proximal skin tissue during the first three days after wounding. (**A**) T-lymphocyte (CD3) IHC staining results. Black arrows indicate infiltrating T-lymphocytes. Magnification is indicated in the figure. Statistical analysis of the number of wound-infiltrating T-lymphocytes (objective 40×). (**B**) B-lymphocytes (CD3) IHC staining results. Black arrows indicate infiltrating B-lymphocytes. Magnification is indicated in the figure. Statistical analysis of the number of wound-infiltrating B-lymphocytes (objective 40×). **wb**: wound bed, **b**: basal layer, **d**: dermis, **sm**: smooth muscle, **ife**: epidermis. Data were analyzed via two-way ANOVA and plotted as mean ± SEM. * *p* < 0.05.

**Figure 3 pharmaceuticals-16-00885-f003:**
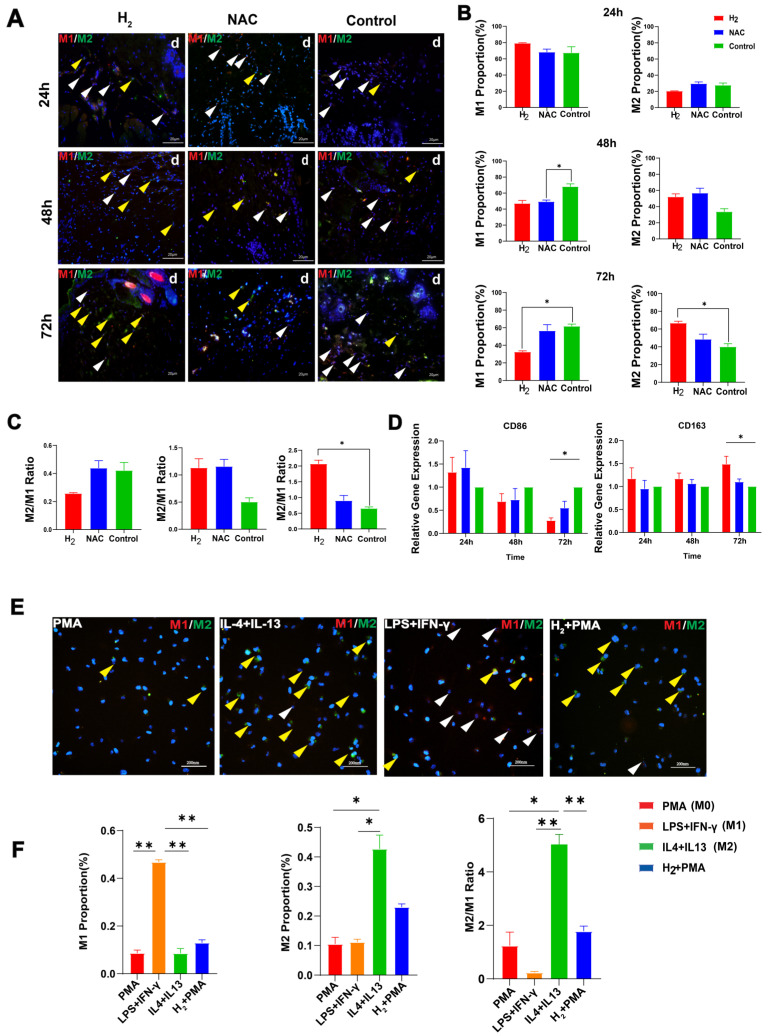
In vivo H_2_ promotes early M2 macrophage polarization during wound healing. In vitro H_2_ induces polarization of THP-1 to M2-type macrophages. (**A**) Representative fluorescence images of specifically stained M1 (CD86, red) and M2 (CD163, green) macrophages in skin tissue at 24 h, 48 h, and 72 h post wounding (yellow arrows point to M2-type macrophages and white arrows point to M1-type macrophages). (**B**) Percentage of M1/M2-type macrophages in each group. (**C**) The ratio of the number of M2/M1 cells in each group at each time point. (**D**) The expression level of M1 (CD86) and M2 (CD163) macrophages marks genes in skin tissue. Data in panels (**B**–**E**) were processed via one-way ANOVA and plotted as mean ± SEM. * *p* < 0.05. (**E**) PMA: THP-1 was induced by adding PMA (100 ng/mL) for 48 h to obtain M0 macrophages. LPS+IFN-γ: M1 macrophages were obtained by adding LPS (100 ng/mL) and IFN-γ (20 ng/mL) to M0 macrophages after 48 h of culture. IL-4 + IL-13: M0 macrophages were incubated with IL-4 (20 ng/mL) and IL-13 (20 ng/mL) for 48 h to obtain M2-type macrophages. H_2_ + PMA: H_2_-containing medium induces M0 polarization (yellow arrows point to M2-type macrophages, white arrows point to M1-type macrophages). (**F**) Histogram statistics of the proportions of M1 and M2 macrophages under each induction method. Data in panel (**B**) were processed via one-way ANOVA and plotted as mean ± SEM. * *p* < 0.05; ** *p* < 0.01.

**Figure 4 pharmaceuticals-16-00885-f004:**
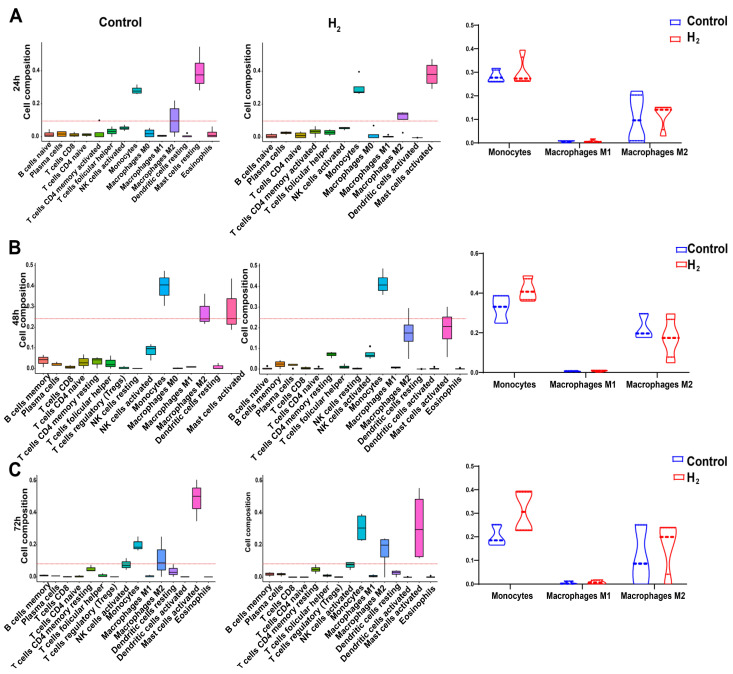
Analysis of immune cell subsets based on expression profile analysis. (**A**–**C**) Boxplots of expression levels of 22 immune cell subpopulations typed from transcriptomic data (24 h, 48 h, 72 h) in the H_2_ and control groups. The right panel represents the comparison of the expression levels of monocytes and macrophages at the three time points.

**Figure 5 pharmaceuticals-16-00885-f005:**
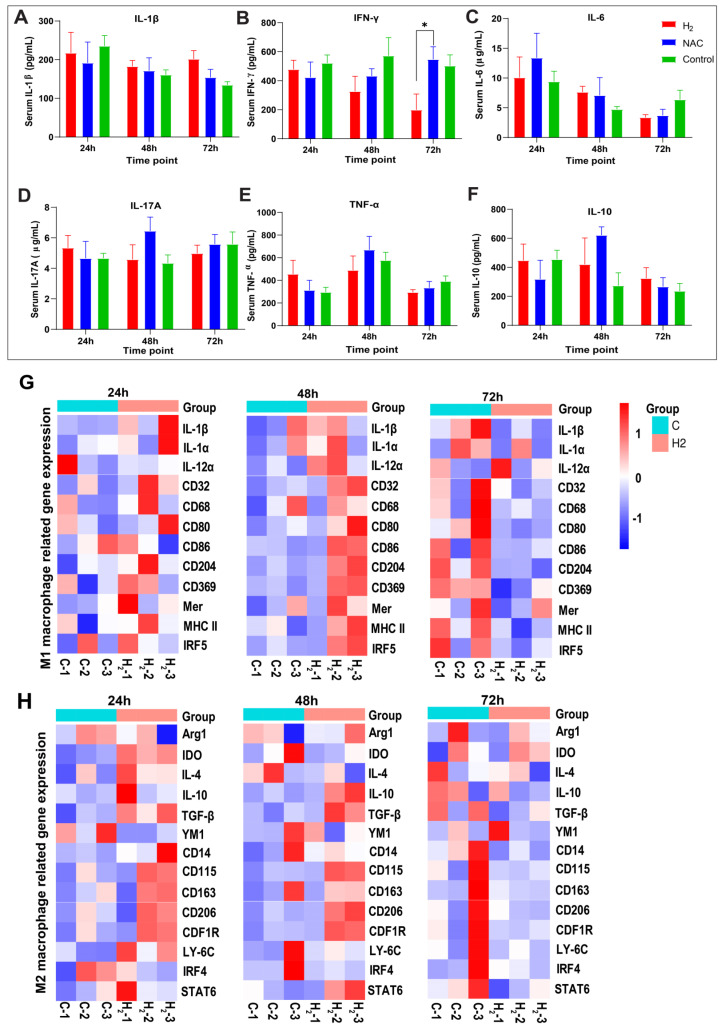
Serological cytokine profiles of the H_2_, NAC, and control groups at 24 h, 48 h, and 72 h post wounding and heatmaps of M1/M2 macrophage-related gene expression. (**A**–**F**) Cytokine profile of IL-1β, IFN-γ, IL-6, IL-17A, TNF-α, and IL-10, respectively. (**G**) Heatmap of genes encoding M1 macrophage-associated secreted proteins, surface-expressed proteins, and transcription factors. (**H**) Heatmap of genes encoding M2 macrophage-associated secreted proteins, surface-expressed proteins, and transcription factors. All data were processed via two−way ANOVA and were plotted as mean ± SEM. * *p* < 0.05; no stars, *p* > 0.05.

**Figure 6 pharmaceuticals-16-00885-f006:**
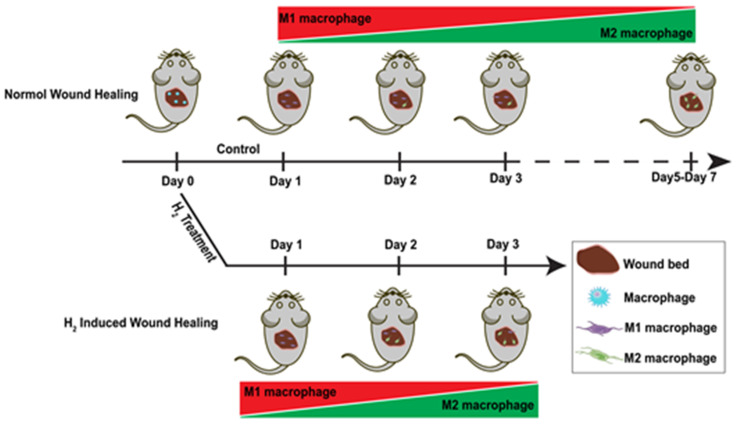
Macrophage polarization processes in normal and H_2_-treated wounds. Day 0 represents the initial wound formation with unpolarized macrophages colonizing the wound. From day 1 to day 3, both wound beds contain mainly M1-type macrophages. A small number of M2-type macrophages appear in normal wounds from day 2, and M2-type macrophages occupy the wound site on days 5–7. In contrast, H_2_-treated wounds had M2-type macrophages occupying the wound as the predominant cell type on day 3.

## Data Availability

Time series analysis of RNAseq Data is contained within the Appendix A. The other datasets used and/or analyzed during the current study are available from the corresponding author on reasonable request.

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
