# Peer review of "Hydrogen Attenuates Inflammation by Inducing Early M2 Macrophage Polarization in Skin Wound Healing"

_pharmaceuticals, 2023, doi:10.3390/ph16060885_

Round 1
Reviewer 1 Report (Previous Reviewer 1)
The authors have introduced most of the suggested changes
Reviewer 2 Report (Previous Reviewer 4)
The manuscript entitle “Hydrogen attenuates inflammation by inducing early M2 mac-2 rophage polarization in skin wound healing” was revised and the version 2 revealed an improved form.
I have few suggestions:
- The authors should pay more attention to the organization of headings subordinate to primary headings;
- A final English spelling and grammar of the text!
In conclusion, the manuscript could be published in the presented revised form.
The manuscript entitle “Hydrogen attenuates inflammation by inducing early M2 mac-2 rophage polarization in skin wound healing” was revised and the version 2 revealed an improved form.
I have few suggestions:
- The authors should pay more attention to the organization of headings subordinate to primary headings;
- A final English spelling and grammar of the text!
In conclusion, the manuscript could be published in the presented revised form.
This manuscript is a resubmission of an earlier submission. The following is a list of the peer review reports and author responses from that submission.
Round 1
Reviewer 1 Report
The article presented by Pengxiang Zhao and collaborates, entitled “Hydrogen attenuates inflammation by inducing early M2 mac-2 rophage polarization in wound healing”, is an original article that aimed to explore the effect of H2 on the change in the phenotype of macrophages in the repair of dorsal tissue full-thickness skin defect mouse. The contribution of the manuscript to scientific literature is medium-low.
Major revision:
1. The title must reflect that the study is at the dermatological level
2. Line 66. “Therefore, bringing forward M1-to-M2 polarization without disturbing the inflammatory stage could be a key strategy to promote wound healing”. The authors should better develop and explain this phrase, because it is important in the introduction and implies the objective of the work and the reason for their work.
3. Line from 70 to 96. The introduction of H2 is too long, part of it can be moved to discussion. Because if it is so clear that H2 is beneficial and favors the transition from M1 to M2, what does the work bring new?
4. In the introduction is necessary to mention the objective more clearly. Line 104
5. Line 116. The mouse model of a total dorsal skin defect needs a reference
6. Fig. 2A. Infiltrated cells are not seen. A leukocyte marker should have been used. How do you quantify if there is no marker?
7. Fig 2 and 3. The control group are mice with damage, there is no NO DAMAGE control group, so the statement that leukocytes rise in the three groups cannot be made since we do not have a reference group (line 159). The line 169 “Monocyte counts were higher in the H2 group than in the other two groups at 24 h; at 72 h, monocyte counts in the H2 group were lower than in the other two groups”. There are no significant changes
8. Fig 4 and 5. How the authors quantify M1/M2 macrophages? It is not specified in material and methods. What does the percentage data refer to? Total of macrophages? Manual counting is a subjective process. A more quantitative and objective technique would be necessary, such as M2/M1 markers by PCR or WB. M1/M2 markers must be specified in figure and text.
9. Fig5. 48 h is a very short time for an M0 macrophage to progress to M2. Since really an M0 is more like an M1 (it is being activated)
Minor revision:
1. Line 14, line 50, line 78, line 89, line 98, line 99, line 180 In vivo with italic.
2. Line 16:” performed time-series” double space
Reviewer 2 Report
This work reproduces another paper of some this paper's authors in Inflamm Regen 2023, where the major setup (wounds, H2, hitmap…) was used o tread wounds using H2 dressings. In this one the authors tried to use H2 during 2 h breathing with the same purpose. The results obtained are less convincing than whose published in Inflamm Regen 2023. It looks like the authors the rest results from a bigger experiment. The only convincing result was the overview of the wounds, I am not sure they are from mice treated by breathing H2.
Zhao P, Dang Z, Liu M, Guo D, Luo R, Zhang M, Xie F, Zhang X, Wang Y, Pan S, Ma X. Molecular hydrogen promotes wound healing by inducing early epidermal stem cell proliferation and extracellular matrix deposition. Inflamm Regen. 2023 Mar 28;43(1):22. doi: 10.1186/s41232-023-00271-9.
Pengxiang Zhao, Zisong Cai, Xujuan Zhang, Mengyu Liu, Fei Xie, Ziyi Liu, Shidong Lu and Xuemei Ma

Reviewer 3 Report
The manuscript by Pengxiang et al have prepared Hydrogen attenuates inflammation by inducing early M2 macrophage polarization in wound healing were conducted. Here are some comments which might help to improve the manuscript.
1. The author has mentioned about the “Compliance with ethical standards” but have not provided the ethical approval number. Please provide the ethical approval number.
2. What is the meaning of symbol Mφ.
3. The author is not sure about the H2 induces polarization and to track macrophage lineages. Explain what type studies need to proposed
4. The author should conduct any studies under hypoxia condition for macrophages or other cell lines.
5. For early-stage healing process, The author should conduct study by taking endothelial cell lines like cell migration or tube formations etc.
6. The number of figures in the manuscript is high and some of the figures must be provided as supporting information.
Reviewer 4 Report
This work is devoted to the investigation of the capacity of molecular hydrogen (H2 to induce early M2 macrophages polarization in wound healing.
The time-series experiments performed in vitro and in vivo by the authors revealed that molecular hydrogen could promote M1 to M2 polarization in the very early stage of inflammation without disrupting M1 activitie and that peripheral blood monocytes were a source of H2-induced M2 macrophages
The subject of the manuscript is interesting, overall well written, comprehensive presented and it faces an interesting issue with possible biomedical applications.
Only one observation – the source of THP-1 cell line, LPS, PMA and antibodies used in immunofluorescence investigation should be specified in Material and Methods Section.
In conclusion the manuscript fits with the scope of the journal and the authors have done a good work finally. Based on my comments the manuscript can be published after minor explanations required above.